# The Validity and Reliability of a Real-Time Biofeedback System for Lumbopelvic Control Training in Baseball Players

**DOI:** 10.3390/s24103060

**Published:** 2024-05-11

**Authors:** Shiu-Min Wang, Po-Hsien Jiang, Kuei-Yuan Chan, Wei-Li Hsu

**Affiliations:** 1School and Graduate Institute of Physical Therapy, College of Medicine, National Taiwan University, No. 17, Xuzhou Rd., Zhongzheng Dist., Taipei City 100, Taiwan; d09428002@ntu.edu.tw; 2Department of Mechanical Engineering, College of Engineering, National Taiwan University, No. 1, Sec. 4, Roosevelt Road, Taipei 106, Taiwan; chuangph@solab.me.ntu.edu.tw (P.-H.J.); chanky@ntu.edu.tw (K.-Y.C.); 3Physical Therapy Centre, National Taiwan University Hospital, No. 1, Changde St., Zhongzheng Dist., Taipei City 100, Taiwan

**Keywords:** baseball, lumbopelvic control, inertial measurement units, pelvic angle

## Abstract

Background: This study validates real-time biofeedback for lumbopelvic control training in baseball. The lumbopelvic region is crucial for generating kinetic energy in pitching. Real-time biofeedback enhances training effectiveness and reduces injury risk. The validity and reliability of this system were examined. Purpose: This study was to investigate the validity and reliability of the real-time biofeedback system for lumbopelvic control training. Methods: Twelve baseball players participated in this study, with data collected in two sessions separated by a week. All participants needed to do the lateral slide exercise and single-leg squat exercise in each session. Pelvic angles detected by the real-time biofeedback system were compared to the three-dimensional motion capture system (VICON) during training sessions. Additionally, pelvic angles measured by the biofeedback system were compared between the two training sessions. Results: The real-time biofeedback system exhibited moderate to strong correlations with VICON in both exercises: lateral slide exercise (*r* = 0.66–0.88, *p* < 0.05) and single-leg squat exercise (*r* = 0.70–0.85, *p* < 0.05). Good to excellent reliability was observed between the first and second sessions for both exercises: lateral slide exercise (ICC = 0.76–0.97) and single-leg squat exercise (ICC = 0.79–0.90). Conclusions: The real-time biofeedback system for lumbopelvic control training, accurately providing the correct pelvic angle during training, could enhance training effectiveness.

## 1. Introduction

In baseball players, the lumbopelvic region contributes significantly (from 50% to 55%) to kinetic energy and force generation during the pitching motion [1,2]. Emphasizing the importance of lumbopelvic control training for baseball players is essential. Previous studies have provided evidence that underscores the important role of gluteal muscles in regulating pelvic and torso movements during the pitching motion [3,4]. Given the direct influence of these muscles on pelvic stability, it is advisable for baseball players to incorporate exercises targeting lumbopelvic control training into their training routine [4]. Moreover, emphasizing the strengthening of the gluteus maximus (GM) through exercises like hip extensions is essential for enhancing lumbopelvic control. Previous studies have mentioned exercises such as the lateral slide exercise and the single-leg squat exercise, which can effectively train the GM [4,5]. These exercises are designed to enhance the athletes’ ability to control their lumbopelvic region during pitching. However, these exercises lack real-time biofeedback to inform players about the correctness of their movements. Inadequate biofeedback may diminish training effectiveness and increase the risk of injury.

To improve training outcomes, previous studies have suggested that real-time biofeedback can help athletes understand their biomechanics, enabling them to perform the correct movements [6,7]. Real-time biofeedback is a concept that can contribute to enhanced performance and a competitive advantage in sports [8,9]. This method uses sensors to assess athletes’ kinematic data, including joint positions and angles that are challenging for a human to perceive. It then displays the processed data back to the athletes for their assessment. In response to this feedback, athletes can make real-time adjustments to their movements to align them with the desired biomechanical parameters, thus completing a feedback loop. Therefore, providing baseball players with real-time biofeedback during their lumbopelvic control training could enhance their biomechanics and performance [10].

It is essential to ensure that athletes execute their movements accurately in order to achieve optimal training results. Real-time biofeedback during lumbopelvic control training, such as the lateral slide and single-leg squat, can provide players with immediate feedback on their movement precision. This biofeedback allows athletes to make timely adjustments and corrections to their lumbopelvic control movements, significantly reducing the risk of injury and enhancing overall lumbopelvic control. To meet these needs, we have developed a real-time biofeedback system explicitly designed for lumbopelvic control training. Our system stands out for its ability to facilitate group training instead of solely concentrating on individual training methods seen in previous equipment [11]. In baseball training, group training is a crucial aspect of school settings. However, conventional training methods cannot deliver instantaneous feedback on players’ adherence to their training regimen. This is where our system comes into play—it equips coaches and trainers with valuable insights into each player’s movements, thereby enhancing training efficiency.

This development combines real-time biofeedback with targeted lumbopelvic exercises, providing baseball players with immediate visual guidance during their training. Participants securely attach an inertial measurement unit (IMU) to an elastic waist belt, covering the anterior and posterior superior iliac spines. Real-time visual biofeedback is displayed on a screen in front of them, facilitating precise adjustments and improvements in lumbopelvic control. However, the validity and reliability of the real-time biofeedback system for lumbopelvic control training have not been investigated before. Therefore, the purpose of this study was to investigate the validity and reliability of the real-time biofeedback system for lumbopelvic control training. It was hypothesized that this system possessed good validity and excellent reliability.

## 2. Materials and Methods

### 2.1. Participants

To determine the appropriate sample size for our study, we utilized GPower 3.1 software. The initial pilot data included 6 participants. The sample size of 10 was calculated based on the pilot data. The parameters used for calculation were a target power of 0.80, an alpha level of 0.05, and an effect size of 0.66.

Twelve college baseball players were recruited for this study. The demographic information of the participants is shown in Table 1. The inclusion criteria were as follows: (1) age between 20 and 40 years; (2) absence of symptoms such as pain, muscle weakness, or numbness. The exclusion criteria were as follows: (1) history of elbow or shoulder surgery; (2) musculoskeletal disorders of the limbs and lumbopelvic within 3 months; and (3) non-compliant individuals. Eligible participants were provided with informed consent and received an explanation of the study procedure before enrolling.

### 2.2. Experimental Procedures

The data collection process involved two sessions, with the second session scheduled one week after the first session. During these sessions, participants performed training movements using the real-time biofeedback system for lumbopelvic control training. Simultaneously, the pelvic angles were captured by the real-time biofeedback system and the 3-dimensional motion capture system. Prior to the actual data collection, participants underwent 5 min of practice trials. Each participant’s entire data collection session lasted approximately 30 min, and they were permitted to take breaks as needed throughout the session.

### 2.3. Real-Time Biofeedback System for Lumbopelvic Control

In this study, we introduce a real-time biofeedback system for lumbopelvic control designed to enhance lower limb control in baseball players. Our system combines back-end data collection, computational analysis, and a user-friendly front-end interface. The interface provides instantaneous feedback to both users and coaches, enabling targeted training exercises to optimize lower body control.

We utilize the Xsens Awinda Inertial Measurement Unit (IMU) system and our custom-developed software, iTraining 1.0.0, to collect data at a 100 Hz sampling rate. This setup enables real-time streaming of angular displacement data from the IMU’s triaxial sensors: a 3D accelerometer (scale: ±160 m/s^2^), a 3D gyroscope (scale: ±2000 deg/s), and a 3D magnetometer (scale: ±1.9 Gauss).

The system architecture comprises a C++ back-end for data collection and analysis and a C# front-end, as depicted in Figure 1, displaying real-time 3-axis angle information.

The real-time biofeedback system using the Xsens Awinda IMU for lumbopelvic control training conducts data acquisition through the C++ back-end. The IMU captures pelvic angle data, including anterior–posterior tilt (AP), upward–downward obliquity (UD), and internal–external rotation (IE) values. These motion parameters are crucial for analyzing and providing real-time feedback during training sessions.

Operators can monitor users’ progress, set specific angular boundary values for different exercises, and provide real-time feedback through intuitive visual cues if movements deviate from predefined angles. The functions of the buttons on the interface are detailed in Table 2.

Central to our system is the integration of real-time biofeedback with lumbopelvic control training principles. Coaches and trainers can tailor pelvic angle constraints for various scenarios, offering athletes immediate visual feedback during exercises and issuing warnings if movements exceed established thresholds. The system tracks and visually represents pelvic tilt angles during exercises, empowering athletes to adjust their movements in real time for enhanced training precision.

Participants securely attach an IMU to an elastic waist belt, covering the anterior superior iliac spines (ASIS) and posterior superior iliac spines (PSIS), as shown in Figure 2 [4,12]. The elastic waist belt covers the entire pelvic region, allowing for the detection of pelvic movements. They can also view real-time biofeedback displayed on the screen in front of them, as depicted in Figure 3.

The real-time biofeedback system for lumbopelvic control training was equipped with an IMU that collected pelvic angle data from the participants at a sampling rate of 100 Hz. A customized MATLAB R2020a software (MathWorks, Natick, MA, USA) was used to determine the range of anterior–posterior tilt (AP), upward–downward obliquity (UD), and internal–external rotation (IE), including the minimum and maximum values of the pelvic angles [13]. The joint kinematics data were filtered using a 2nd-order low-pass Butterworth filter with a 10 Hz cut-off frequency [14].

The system incorporates two specific training movements: (1) lateral slide exercise; (2) single-leg squat exercise.

Lateral Slide Exercise

This exercise required participants to execute a single-leg squat with a lateral slide while keeping the test knee flexed at 90 degrees and the trunk in an upright position, as shown in Figure 4a. Participants were instructed to cross their arms over their chest. During the warm-up phase, the maximum distance achievable for outward sliding was measured and designated as the target distance. The motion limited the degree of upward–downward pelvic tilt, requiring participants to uphold an upright posture while sliding the non-test leg laterally to attain the target distance.

For each leg, two training trials were conducted. Participants were enabled to modify their trunk angle with real-time visual biofeedback.

Single-Leg Squat Exercise

The exercise required participants to perform squats while simultaneously maintaining balance and an upright torso position. Additionally, participants needed to cross their arms over their chest and flex the knee of the non-testing leg to a 90-degree angle, as shown in Figure 4b. The motion limited the extent of anterior–posterior pelvic tilt, requiring participants to maintain an upright posture throughout.

For each leg, two training trials were conducted. Participants were enabled to modify their pelvic angle with real-time visual biofeedback.

### 2.4. A 3-Dimensional Motion Capture System

A 3-dimensional motion capture system (VICON ver. 2.5, Oxford Metrics Ltd., Oxford, UK) was employed. It was equipped with ten infrared cameras (VICON Bonita, Oxford Metrics, UK) to capture joint kinematic data during lumbopelvic control training movements. These data were sampled at a rate of 120 Hz. Forty-five spherical retro-reflective markers (14 mm in diameter) were used, strategically positioned according to the Plug-in-Gait model’s anatomical landmarks [15,16]. Our chosen motion analysis system was VICON, renowned as the golden standard [17]. It uses infrared detection to track the position of reflective markers accurately. The pelvic angles were calculated using the 3-dimensional motion capture system. A customized MATLAB R2020a software (MathWorks, Natick, MA, USA) was used to determine the range of anterior–posterior tilt (AP), upward–downward obliquity (UD), and internal–external rotation (IE), including the minimum and maximum values of the pelvic angles [13]. The joint kinematics data were filtered using a 2nd-order low-pass Butterworth filter with a 10 Hz cut-off frequency [14].

### 2.5. Statistical Analysis

PASW Statistics 18.0 for Windows (SPSS, Chicago, IL, USA) was used to do the statistical analysis for this study.

A Shapiro–Wilk test was employed to assess the normality of the data. As all the data exhibited a normal distribution, parametric methods were utilized. To assess the validity of the real-time biofeedback system for lumbopelvic control training, we employed the paired samples t-test to evaluate systematic bias between sessions and tools. Pearson product–moment correlation was used, and the correlation strength (represented as ‘r’) was categorized as weak (0 to 0.49), moderate (0.50 to 0.75), and strong (>0.75) [18]. The standardized differences in means, along with 95% confidence intervals (CIs), were also calculated to determine the magnitude of change across and between tests. Cohen’s d effect size (ES) was used to classify magnitudes of change as trivial (<0.2), small (0.2 to 0.49), moderate (0.5 to 0.79), large (0.8 to 1.60), and very large (>1.60) [19]. Bland–Altman analysis with 95% limits of agreement was also calculated.

To analyze the reliability between test–retest, the intraclass correlation coefficient (ICC) was used. The minimum and maximum values of pelvic tilt angle for each subject on two testing sessions were retrieved from the training system. The interpretation of ICC values was categorized as poor (<0.5), moderate (0.5 to 0.75), good (0.76 to 0.9), and excellent (>0.9) [20,21].

## 3. Results

### 3.1. Validity of a Real-Time Biofeedback System

In the lateral slide exercise, no significant differences (*p* > 0.05) were found between the real-time biofeedback system and VICON for all tilt angle directions, with trivial to moderate effect sizes, i.e., AP angle: 0.57, UD angle: 0.13, and IE angle: 0.16. Moderate to strong correlations were observed between the real-time biofeedback system and VICON for the AP angle (*r* = 0.85, *p* < 0.01), UD angle (*r* = 0.66, *p* = 0.02), and IE angle (*r* = 0.88, *p* < 0.01) (Table 2). The Bland–Altman plot between the real-time biofeedback system and VICON for the three tilt angles in the lateral slide exercise is shown in Figure 5. The figures revealed that 0/12 (0%), 1/12 (8.33%), and 0/12 (0%) of the data points fell beyond the mean ± 1.96 SD lines for the three tilt angle directions, i.e., AP, UD, and IE, respectively. The Bland–Altman plot illustrated that the majority of data points were within the 95% confidence intervals. The mean bias between the two systems was 0.48 degrees for the AP angle, −0.40 degrees for the UD angle, and 0.58 degrees for the IE angle.

In the single-leg squat exercise, no significant differences (*p* > 0.05) were found between the real-time biofeedback system and VICON for all tilt angle directions, with trivial to small effect sizes, i.e., AP angle: 0.39, UD angle: 0.42, and IE angle: 0.35. Moderate to strong correlations were observed between the real-time biofeedback system and VICON for the AP angle (*r* = 0.70, *p* = 0.01), UD angle (*r* = 0.85, *p* < 0.01), and IE angle (*r* = 0.79, *p* < 0.01) (Table 3). The Bland–Altman plot between the real-time biofeedback system and VICON for the three tilt angles in the single-leg squat is shown in Figure 6. The figures revealed that 1/12 (8.33%), 1/12 (8.33%), and 0/12 (0%) of the data points fell beyond the mean ± 1.96 SD lines for the three tilt angle directions, i.e., AP, UD, and IE, respectively. The Bland–Altman plot illustrated that the majority of data points were within the 95% confidence intervals. The mean bias between the two systems was 0.91 degrees for the AP angle, −0.94 degrees for the UD angle, and 0.83 degrees for the IE angle.

### 3.2. Reliability of a Real-Time Biofeedback System

In the lateral slide exercise, the real-time biofeedback system showed good to excellent reliability between the first and second sessions at the AP angle (ICC = 0.76), UD angle (ICC = 0.97), and IE angle (ICC = 0.84) (Table 4).

In the single-leg squat exercise, the real-time biofeedback system showed good to excellent reliability between the first and second sessions at the AP angle (ICC = 0.85), UD angle (ICC = 0.79), and IE angle (ICC = 0.90), as shown in Table 4.

## 4. Discussion

The purpose of this study was to investigate the validity and reliability of the real-time biofeedback system for lumbopelvic control training. Our findings provide valuable insights into the potential of this system as a tool to enhance the effectiveness of such training programs.

In the validity test, although we employed the Xsens IMU, which has demonstrated high validity [22], there have not been specific studies validating its application in assessing lumbopelvic control movement. Nevertheless, we observed a significant, moderate-to-strong correlation between the real-time biofeedback system and the VICON motion capture system. This correlation suggests that our system accurately captured pelvic angles in various directions, reinforcing its validity in tracking and assessing lumbopelvic control during training sessions. In the real-time biofeedback system, when compared with the AP and IE directions in the lateral slide exercise, the UD direction exhibited a lower correlation with VICON. However, the effect size suggested that the difference in the UD direction was trivial compared to VICON. Moreover, the Bland–Altman plot depicted only one data point outside the 95% confidence intervals. Similarly, in the single-leg squat exercise, the AP angle showed a moderate correlation between the real-time biofeedback system and the VICON motion capture system, with the effect size indicating a small difference compared to VICON. The Bland–Altman plot for this exercise also displayed one data point outside the 95% confidence intervals. Thus, despite slightly lower correlations in the UD direction for the lateral slide exercise and the AP direction for the single-leg squat exercise, the real-time biofeedback system still demonstrates good validity. Compared to previous studies that used different IMUs to detect pelvic motion, our study demonstrated a higher correlation with the golden standard motion capture system than those particular IMUs. Previous studies indicated their IMUs did not correlate significantly with the golden standard motion capture system in AP and UD directions [23,24]. Therefore, our system exhibits superior validity in assessing pelvic motion compared to the IMUs used in the previous studies.

In the reliability test, our findings indicated good to excellent reliability for the real-time biofeedback system in both the lateral slide and single-leg squat exercises across two different sessions. Specifically, in the lateral slide exercise, the UD direction of the pelvic angle, which had limitations imposed, displayed excellent reliability. Likewise, in the single-leg squat exercise, the AP direction of the pelvic angle with imposed limitation angles showed good reliability across both sessions. These outcomes underscore the consistency and dependability of our system, making it suitable for continuous monitoring of pelvic angles over time.

One notable aspect of our study is the integration of real-time biofeedback into lumbopelvic control training for baseball players. Various training programs, such as the single-leg bridge maneuver [4], single-leg squat, lateral slide exercise [25], and others, have been devised and incorporated to enhance lumbopelvic control. However, these programs frequently lack real-time biofeedback to provide players with immediate guidance on their movement execution. Our research demonstrates the potential benefits of incorporating real-time biofeedback into these training exercises. Immediate feedback allows athletes to make real-time adjustments, which can lead to more effective training and potentially reduce the risk of injury. This highlights the advantage of our developed system in improving training outcomes and enhancing athletes’ ability to control their lumbopelvic region.

Despite the promising results, our study does have limitations. We were able to control the slide distance in the lateral slide exercise during training, but we were unable to standardize the squat height for the single-leg squat exercise. Specifically, the instruction for the single-leg squat exercise was to flex the knee of the testing leg to a 90-degree angle. However, we could not confirm whether the subjects consistently achieved this exact 90-degree angle in each training session. This variability in knee flexion may have had an impact on the results.

## 5. Conclusions

The findings of this study have significant clinical implications for the field of baseball training. The development and validation of a real-time biofeedback system for lumbopelvic control training, utilizing IMU, offers a valuable tool to enhance the effectiveness of lumbopelvic control exercises. The strong correlation with VICON and the excellent reliability show that this system can provide immediate guidance to athletes during lumbopelvic control training, helping them improve their lumbopelvic control. This technology can not only aid baseball pitchers in optimizing their movement patterns but also has the potential to be applied in various rehabilitation and sports training programs.

This study emphasizes the potential impact of a real-time biofeedback system on lumbar pelvic control training for baseball players. Our system provides immediate feedback during exercises designed to enhance lumbopelvic control, significantly contributing to improved training outcomes. In our upcoming research, we plan to leverage this technology to enhance athletes’ lumbar pelvic motion control, aiming to assess its effectiveness in reducing the risk of sports injuries resulting from incorrect movements. Our overarching goal is to not only optimize training effectiveness for players but also cultivate a safer sports culture. To achieve this, we have implemented an angle range within the training regimen, tailored to meet the specific requirements set by trainers. If the angle surpasses this predefined range during training, a warning prompt is displayed, prompting players to adjust their movements in real time based on feedback values.

## Figures and Tables

**Figure 1 sensors-24-03060-f001:**
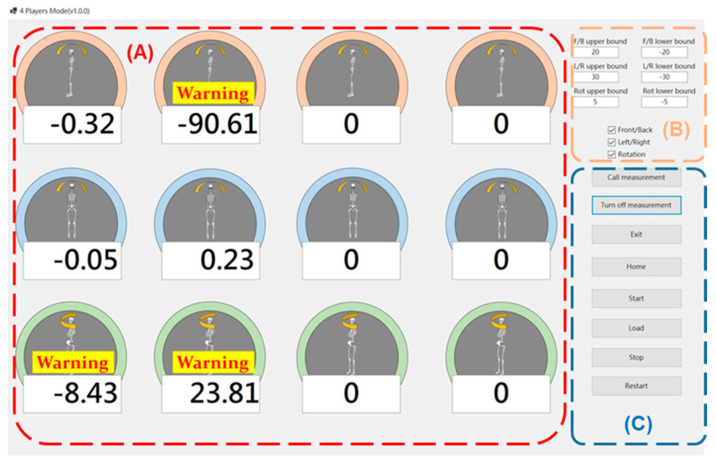
The interface of the self-developed software. Panel (**A**) displays the real-time pelvic angle in anterior–posterior tilt, right–left tilt, and right-left rotation. Panel (**B**) sets limitations on pelvic angles during lumbopelvic control movements. Panel (**C**) buttons for controlling the application.

**Figure 2 sensors-24-03060-f002:**
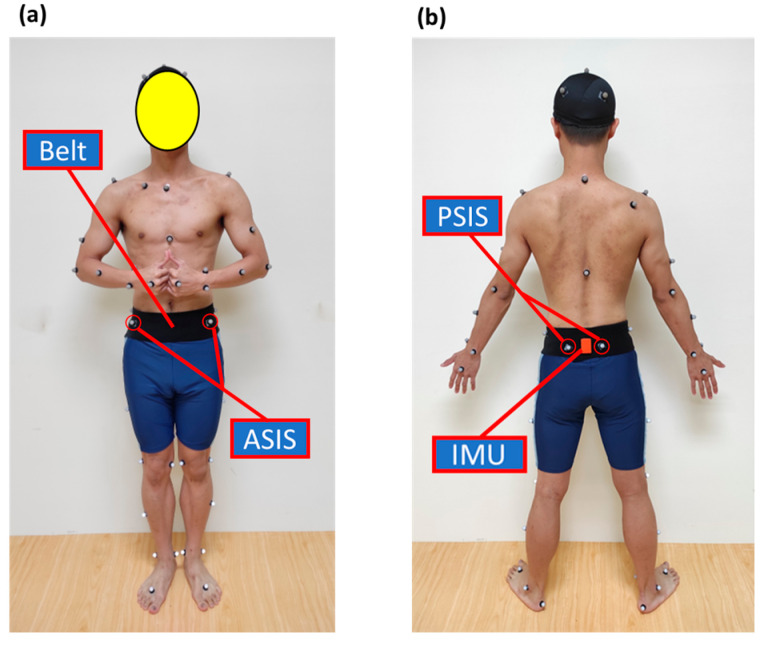
Placement of IMU covering the anterior superior iliac spines (ASIS) and posterior superior iliac spines (PSIS) in the real-time biofeedback system for lumbopelvic control training. (**a**) Anterior view; (**b**) posterior view.

**Figure 3 sensors-24-03060-f003:**
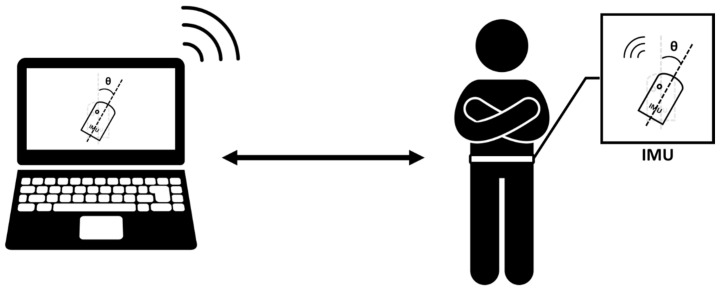
Schematic representation of a real-time biofeedback system for lumbopelvic control training. The IMU was attached to a belt covering the anterior and posterior superior iliac spines. Real-time visual biofeedback is displayed on a screen in front of the subject.

**Figure 4 sensors-24-03060-f004:**
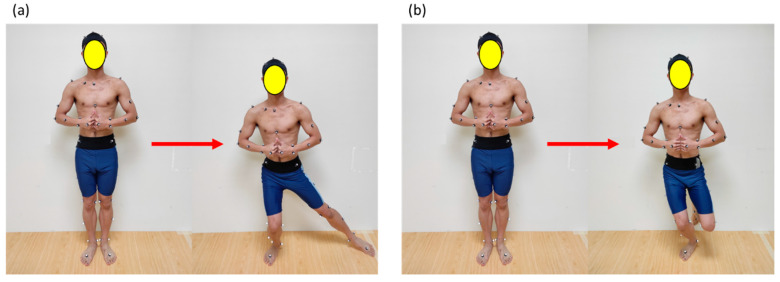
Lumbopelvic control movements: (**a**) lateral slide exercise: participants perform a single-leg squat with a lateral slide, keeping one knee bent at 90 degrees and their upper body straight; (**b**) single-leg squat exercise: participants do squats while balancing, crossing their arms over their chest, and bending the other knee to 90 degrees.

**Figure 5 sensors-24-03060-f005:**
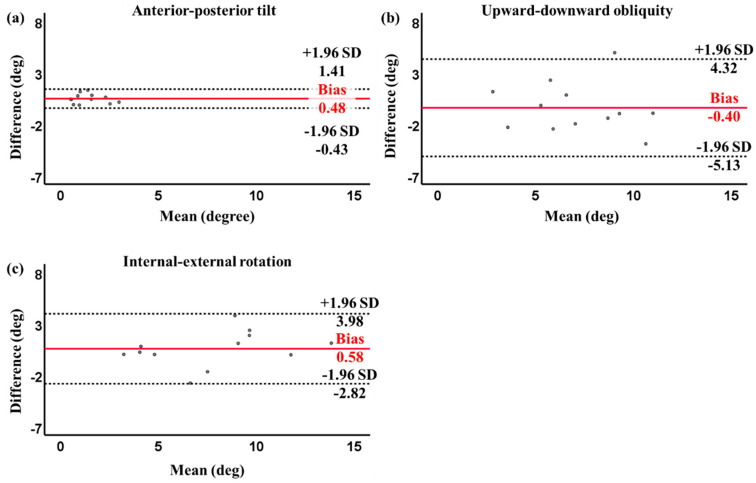
Bland–Altman plot of pelvic angles with 95% limits of agreement (dashed lines) and the mean difference (solid red line) between the real-time biofeedback system and VICON for lateral slide exercise: (**a**) anterior–posterior tilt; (**b**) upward–downward obliquity; and (**c**) internal–external rotation. Difference: real-time biofeedback system − VICON. Mean: (real-time biofeedback system + VICON)/2.

**Figure 6 sensors-24-03060-f006:**
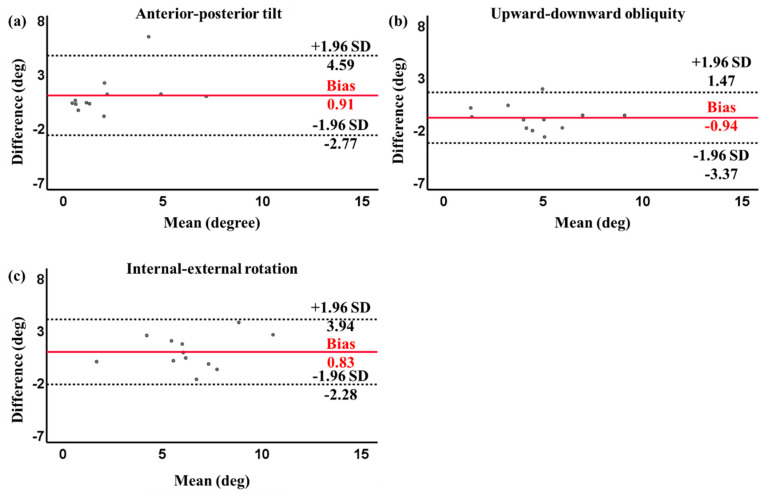
Bland–Altman plot of pelvic angles with 95% limits of agreement (dashed lines) and the mean difference (solid red line) between the real-time biofeedback system and VICON for single-leg squat exercise: (**a**) anterior–posterior tilt; (**b**) upward–downward obliquity; and (**c**) internal–external rotation. Difference: real-time biofeedback system − VICON. Mean: (real-time biofeedback system + VICON)/2.

**Table 1 sensors-24-03060-t001:** Demographic information of participants (number of subjects = 12).

Demographic Information	Mean (±SD)
Gender (male/female)	12/0
Age (years)	23.42 (1.83)
Height (cm)	172.25 (4.75)
Mass (kg)	66.58 (10.68)
BMI (kg/m^2^)	22.44 (3.61)

SD: standard deviation; BMI: body mass index.

**Table 2 sensors-24-03060-t002:** Function of the buttons on the interface.

Button Name	Function
Call Measurement	Initiate the back-end C++ program to establish a connection with the IMU sensor data.
Turn-off Measurement	Terminate the back-end C++ program.
Exit	Shut down the real-time biofeedback system.
Home	Return to the system’s main interface.
Start	Analyze the sensor data to calculate the initial value resets and compute the angle variations.
Load	Load the data into the front-end visualization interface.
Stop	Cease the angle computation.
Restart	Restart the computation of angles.

**Table 3 sensors-24-03060-t003:** Pearson’s correlation coefficients of a real-time biofeedback system for lumbopelvic control training and VICON pelvic angle measurements.

Movement	Pelvic Angle	*r*	95%CI	*p*
Lateral slide	AP	0.85	(0.19~0.79)	<0.01 *
	UD	0.66	(−1.94~1.13)	0.02 *
	IE	0.88	(−0.52~1.68)	<0.01 *
Single-leg squat	AP	0.70	(−0.28~2.11)	0.01 *
	UD	0.85	(−1.73~−0.16)	<0.01 *
	IE	0.79	(−1.77~1.82)	<0.01 *

* Statistically significant correlation (*p* < 0.05); AP: anterior–posterior tilt; UD: upward–downward obliquity; IE: internal–external rotation.

**Table 4 sensors-24-03060-t004:** Test–retest reliability of a real-time biofeedback system for lumbopelvic control training pelvic angle measurements.

Movement	Pelvic Angle	ICC	95% CI	*p*
Lateral slide	AP	0.76	(0.17~0.93)	0.01 *
	UD	0.97	(0.89~0.99)	<0.01 *
	IE	0.84	(0.45~0.95)	<0.01 *
Single-leg squat	AP	0.85	(0.49~0.95)	<0.01 *
	UD	0.79	(0.24~0.94)	0.01 *
	IE	0.90	(0.67~0.97)	<0.01 *

* Statistically significant correlation (*p* < 0.05); AP: anterior–posterior tilt; UD: upward–downward obliquity; IE: internal–external rotation.

## Data Availability

The original contributions presented in the study are included in the article, further inquiries can be directed to the corresponding author.

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
