# Peer review of "The Validity and Reliability of a Real-Time Biofeedback System for Lumbopelvic Control Training in Baseball Players"

_sensors, 2024, doi:10.3390/s24103060_

Round 1

Reviewer 1 Report

Comments and Suggestions for Authors

In this work, the authors showed a biofeedback system for lumbopelvic control training and investigated its validity and reliability. Some parameters related to the performance of the system were calculated. From perspective of the product and practical clinical application value, the lack of the number of subjects was striking and the experimental data collection time seemed insufficient. Too many similar researches could be found respect to real-time biofeedback, while no impressive innovation was proposed in this manuscript. Improvement on text clarity, completeness of experiments, and the correlation of the system functionalities are required for a publication-deserving manuscript. Some concerns are listed as follows:

(1) There are many mature real-time biofeedback-related products on the market, such as BiosignalsPLUX. What is the difference between the system in the manuscript and these similar products?

(2) If possible, more participants should be invited for the experiment and the time of data collection should be more reasonable.

(3) In figure 1, only software interface was shown. How does the system work? The authors should provide images of the software in action and more details in the text.

(4) The authors should add some text annotations in figures 2 and 4.

(5) The text in 2.3. Real-Time Biofeedback System for Lumbopelvic Control is unclear. The authors should improve its readability.

(6) Page 5-6, “Data from the motion capture system was processed with a custom MATLAB R2020a program”, how was the data processed? The authors should provide more details about data processing procedure.

Comments on the Quality of English Language

None

Author Response

Comments from Reviewer 1:

In this work, the authors showed a biofeedback system for lumbopelvic control training and investigated its validity and reliability. Some parameters related to the performance of the system were calculated. From perspective of the product and practical clinical application value, the lack of the number of subjects was striking and the experimental data collection time seemed insufficient. Too many similar researches could be found respect to real-time biofeedback, while no impressive innovation was proposed in this manuscript. Improvement on text clarity, completeness of experiments, and the correlation of the system functionalities are required for a publication-deserving manuscript. Some concerns are listed as follows:

(1) There are many mature real-time biofeedback-related products on the market, such as BiosignalsPLUX. What is the difference between the system in the manuscript and these similar products?

Response:

We thank Reviewer 1 for the recommendations. According to the suggestions, we have added a description of the difference between the system and mature real-time biofeedback-related products in the revised manuscript. The revised contents are as follows:

(1)    In 1. Introduction page 2 lines 62 to 67

Our system stands out for its ability to facilitate group training instead of solely concentrating on individual training methods seen in previous equipment. In baseball training, group training is a crucial aspect of school settings. However, conventional training techniques cannot deliver instantaneous feedback on players' adherence to their training regimen. This is where our system comes into play - it equips coaches and trainers with valuable insights into each player's movements, thereby enhancing training efficiency.

(2)    In 4. Discussion page 11 lines 291 to 297

Compared to previous studies that used different IMUs to detect pelvic motion, our study demonstrated a higher correlation with the golden standard motion capture system than those particular IMUs. Previous studies indicated their IMUs did not correlate significantly with the golden standard motion capture system in AP and UD directions. Therefore, our system exhibits superior validity in assessing pelvic motion compared to the IMUs used in the previous studies.

(2) If possible, more participants should be invited for the experiment and the time of data collection should be more reasonable.

Response:

We thank Reviewer 1 for the recommendations. The initial pilot data included 6 participants. GPower 3.1 software was used to conduct a power analysis, determining the required sample size. A total of 10 individuals were planned for the study to achieve a target power of 0.80, an alpha level of 0.05, and an effect size of 0.66 based on the pilot data. The current collected sample size has exceeded the initially calculated sample size, and all statistical analyses have shown significant correlations. The revised contents are as follows:

(1)    In 2. Materials and Methods 2.1. Participants page 2 lines 80 to 82

To determine the appropriate sample size for our study, we utilized GPower 3.1 software. Our goal was to include at least 10 individuals with a target power of 0.80, an alpha level of 0.05, and an effect size of 0.66, which we based on our pilot data.

(3) In figure 1, only software interface was shown. How does the system work? The authors should provide images of the software in action and more details in the text.

Response:

We thank Reviewer 1 for the recommendations. According to the suggestions, we have added descriptions and changed Figure 1 to show how the system works in the revised manuscript. Moreover, we added Table 2 to show the function of the system. The revised contents are as follows:

(1)  In Figure 1 page 4 lines 124 to 127

Figure 1. The interface of the self-developed software. Panel (A) displays the real-time pelvic angle in anterior-posterior tilt, right-left tilt, and right-left rotation; Panel (B) sets limitations on pelvic angles during lumbopelvic control movements; Panel (C) buttons for controlling the application.

(2)  In Table 2 page 4 lines 129

Table 2. Function of the buttons on the interface

(3)  In 2. Materials and Methods 2.3. Real-Time Biofeedback System for Lumbopelvic Control page 4 lines 131 to 136

Central to our system is the integration of real-time biofeedback with lumbopelvic control training principles. Coaches and trainers can tailor pelvic angle constraints for various scenarios, offering athletes immediate visual feedback during exercises and issuing warnings if movements exceed established thresholds. The system tracks and visually represents pelvic tilt angles during exercises, empowering athletes to adjust their movements in real time for enhanced training precision.

(4) The authors should add some text annotations in figures 2 and 4.

 Response:

We thank Reviewer 1 for the recommendations. According to the suggestions, we have included detailed descriptions and updated Figure 2 and Figure 4 to illustrate the position of the IMU and the correct method for performing the lumbopelvic control movement in the revised manuscript. The revised contents are as follows:

(1)  In Figure 2 page 5 lines 142 to 144

Figure 2. Placement of IMU covering the anterior superior iliac spines (ASIS) and posterior superior iliac spines (PSIS) in the real-time biofeedback system for lumbopelvic control training; (a) anterior view; (b) posterior view.

(2)  In Figure 4 page 6 lines 178 to 181

Figure 4. Lumbopelvic control movements: (a) lateral slide exercise: participants perform a single-leg squat with a lateral slide, keeping one knee bent at 90 degrees and their upper body straight ;(b) single-leg squat exercise: participants do squats while balancing, crossing their arms over their chest and bending the other knee to 90 degrees.

(5) The text in 3. Real-Time Biofeedback System for Lumbopelvic Controlis unclear. The authors should improve its readability.

Response:

We thank Reviewer 1 for the recommendations. According to the suggestions, we have improved the readability of the text in 2.3. Real-Time Biofeedback System for Lumbopelvic Control in the revised manuscript. The revised contents are as follows:

(1)  In 2. Materials and Methods 2.3. Real-Time Biofeedback System for Lumbopelvic Control page 3 lines 107 to 111

In this study, we introduce a Real-Time Biofeedback System for Lumbopelvic Control, designed to enhance lower limb control in baseball players. Our system combines back-end data collection, computational analysis, and a user-friendly front-end interface. The interface provides instantaneous feedback to both users and coaches, enabling targeted training exercises to optimize lower body control.

(2)  In 2. Materials and Methods 2.3. Real-Time Biofeedback System for Lumbopelvic Control page 3 lines 112 to 116

We utilize the Xsens Awinda Inertial Measurement Unit (IMU) system and our custom-developed software, iTraining 1.0.0, collecting data at a 100 Hz sampling rate. This setup enables real-time streaming of angular displacement data from the IMU's triaxial sensors: a 3D accelerometer (scale: ±160 m/s²), a 3D gyroscope (scale: ±2000 deg/s), and a 3D magnetometer (scale: ±1.9 Gauss).

(3)  In 2. Materials and Methods 2.3. Real-Time Biofeedback System for Lumbopelvic Control page 3 lines 117 to 122

The system architecture comprises a C++ back-end for data collection and analysis, and a C# front-end, as depicted in Figure 1, displaying real-time 3-axis angle information. Operators can monitor users' progress, set specific angular boundary values for different exercises, and provide real-time feedback through intuitive visual cues if movements deviate from predefined angles. The functions of the buttons on the interface are detailed in Table 2.

(4)  In 2. Materials and Methods 2.3. Real-Time Biofeedback System for Lumbopelvic Control page 4 lines 131 to 136

Central to our system is the integration of real-time biofeedback with lumbopelvic control training principles. Coaches and trainers can tailor pelvic angle constraints for various scenarios, offering athletes immediate visual feedback during exercises and issuing warnings if movements exceed established thresholds. The system tracks and visually represents pelvic tilt angles during exercises, empowering athletes to adjust their movements in real time for enhanced training precision.

(5)  In 2. Materials and Methods 2.3. Real-Time Biofeedback System for Lumbopelvic Control page 4 lines 137 to 140

Participants securely attach an IMU to an elastic waist belt, covering the anterior superior iliac spines (ASIS) and posterior superior iliac spines (PSIS), as shown in Figure 2. They can also view real-time biofeedback displayed on the screen in front of them, as depicted in Figure 3.

(6) Page 5-6, “Data from the motion capture system was processed with a custom MATLAB R2020a program”, how was the data processed? The authors should provide more details about data processing procedure.

Response:

We thank Reviewer 1 for the recommendations. According to the suggestions, we have provided more details about the data processing procedure in the revised manuscript. The revised contents are as follows:

(1)  In 2. Materials and Methods 2.3. Real-Time Biofeedback System for Lumbopelvic Control page 6 lines 150 to 156

The real-time biofeedback system for lumbopelvic control training was equipped with an IMU that collected pelvic angle data from the participants at a sampling rate of 100 Hz. A customized MATLAB R2020a software (MathWorks, Natick, MA, USA) was used to determine the range of anterior-posterior tilt (AP), upward-downward obliquity (UD), and internal-external rotation (IE), including the minimum and maximum values of the pelvic angles. The joint kinematics data was filtered using a 2nd-order low-pass Butterworth filter with a 10 Hz cut-off frequency.

(2)  In 2. Materials and Methods 2.4. 3-Dimensional Motion Capture System page 7 lines 190 to 195

The pelvic angles were calculated using the 3-dimensional motion capture system. A customized MATLAB R2020a software (MathWorks, Natick, MA, USA) was used to determine the range of anterior-posterior tilt (AP), upward-downward obliquity (UD), and internal-external rotation (IE), including the minimum and maximum values of the pelvic angles. The joint kinematics data was filtered using a 2nd-order low-pass Butterworth filter with a 10 Hz cut-off frequency.

Reviewer 2 Report

Comments and Suggestions for Authors

The manuscript analyses various aspects of the effectiveness of a system for training lumbopelvic control in athletes. The results of the study will find application in quality and safety control systems for sports activities. The paper describes the main features of the design and development of such systems and validates the results obtained. But the question remains as to what specific improvements were proposed in the study.

It should also be noted that there is no comparison of the results with other developments in this field, including in comparison with existing world analogues (2). I believe that this literature review is incomplete and needs to be improved (1).  

In addition, I would like to propose to consider a similar qualisys motion capture system. This system is one of the leaders in the field of motion analysis, but it is not mentioned in the article.

There is no description of the characteristics of the sensors. Please provide a description of the hardware performance in general (3). Describe the methodology of IMU sensors arrangement(4)

What are the next stages of the research planned? (5)

In addition, I would like to note that the interface shown in Figure 1 is not informative. There is no explanation of the matrix of figures (6) , and the interface is not translated into English (7)

Author Response

Comments from Reviewer 2:

The manuscript analyses various aspects of the effectiveness of a system for training lumbopelvic control in athletes. The results of the study will find application in quality and safety control systems for sports activities. The paper describes the main features of the design and development of such systems and validates the results obtained. But the question remains as to what specific improvements were proposed in the study.

It should also be noted that there is no comparison of the results with other developments in this field, including in comparison with existing world analogues (2). I believe that this literature review is incomplete and needs to be improved (1). 

Response:

We thank Reviewer 2 for the recommendations. According to the suggestions, we have added descriptions of the difference between the system and mature real-time biofeedback-related products in the revised manuscript. The revised contents are as follows:

(1)    In 1. Introduction page 2 lines 62 to 67

Our system stands out for its ability to facilitate group training instead of solely concentrating on individual training methods seen in previous equipment. In baseball training, group training is a crucial aspect of school settings. However, conventional training techniques cannot deliver instantaneous feedback on players' adherence to their training regimen. This is where our system comes into play - it equips coaches and trainers with valuable insights into each player's movements, thereby enhancing training efficiency.

(2)    In 4. Discussion page 11 lines 291 to 297

Compared to previous studies that used different IMUs to detect pelvic motion, our study demonstrated a higher correlation with the golden standard motion capture system than those particular IMUs. Previous studies indicated their IMUs did not correlate significantly with the golden standard motion capture system in AP and UD directions. Therefore, our system exhibits superior validity in assessing pelvic motion compared to the IMUs used in the previous studies.

In addition, I would like to propose to consider a similar qualisys motion capture system. This system is one of the leaders in the field of motion analysis, but it is not mentioned in the article.

Response:

We thank Reviewer 2 for the recommendations. According to the suggestions, we added descriptions of the reasons why we used VICON motion capture system to compare with the real-time biofeedback system, not the Qualisys motion capture system. The revised contents are as follows:

(1)  In 2. Materials and Methods 2.4. 3-Dimensional Motion Capture System page 7 lines 188 to 190

Our chosen motion analysis system was VICON, renowned as the golden standard. It uses infrared detection to track the position of reflective markers accurately.

There is no description of the characteristics of the sensors. Please provide a description of the hardware performance in general (3). Describe the methodology of IMU sensors arrangement(4)

Response:

We thank Reviewer 2 for the recommendations. According to the suggestions, we have added descriptions of the hardware performance and changed Figure 2. We have also added descriptions of the methodology of IMU sensor arrangement in the revised manuscript. The revised contents are as follows:

(1)  In 2. Materials and Methods 2.3. Real-Time Biofeedback System for Lumbopelvic Control page 3 lines 107 to 111

In this study, we introduce a Real-Time Biofeedback System for Lumbopelvic Control, designed to enhance lower limb control in baseball players. Our system combines back-end data collection, computational analysis, and a user-friendly front-end interface. The interface provides instantaneous feedback to both users and coaches, enabling targeted training exercises to optimize lower body control.

(2)  In 2. Materials and Methods 2.3. Real-Time Biofeedback System for Lumbopelvic Control page 3 lines 112 to 116

We utilize the Xsens Awinda Inertial Measurement Unit (IMU) system and our custom-developed software, iTraining 1.0.0, collecting data at a 100 Hz sampling rate. This setup enables real-time streaming of angular displacement data from the IMU's triaxial sensors: a 3D accelerometer (scale: ±160 m/s²), a 3D gyroscope (scale: ±2000 deg/s), and a 3D magnetometer (scale: ±1.9 Gauss).

(3)  In 2. Materials and Methods 2.3. Real-Time Biofeedback System for Lumbopelvic Control page 3 lines 117 to 122

The system architecture comprises a C++ back-end for data collection and analysis, and a C# front-end, as depicted in Figure 1, displaying real-time 3-axis angle information. Operators can monitor users' progress, set specific angular boundary values for different exercises, and provide real-time feedback through intuitive visual cues if movements deviate from predefined angles. The functions of the buttons on the interface are detailed in Table 2.

(4)  In 2. Materials and Methods 2.3. Real-Time Biofeedback System for Lumbopelvic Control page 4 lines 131 to 136

Central to our system is the integration of real-time biofeedback with lumbopelvic control training principles. Coaches and trainers can tailor pelvic angle constraints for various scenarios, offering athletes immediate visual feedback during exercises and issuing warnings if movements exceed established thresholds. The system tracks and visually represents pelvic tilt angles during exercises, empowering athletes to adjust their movements in real time for enhanced training precision.

(5)  In 2. Materials and Methods 2.3. Real-Time Biofeedback System for Lumbopelvic Control page 4 lines 137 to 140

Participants securely attach an IMU to an elastic waist belt, covering the anterior superior iliac spines (ASIS) and posterior superior iliac spines (PSIS), as shown in Figure 2. They can also view real-time biofeedback displayed on the screen in front of them, as depicted in Figure 3.

What are the next stages of the research planned? (5)

Response:

We thank Reviewer 2 for the recommendations. According to the suggestions, we have added a description in the revised manuscript to show the next research stages. The revised contents are as follows:

(1)  In 5. Conclusions page 12 lines 336 to 340

Our upcoming research will utilize this training equipment to improve athletes' lumbar pelvic motion control. We aim to determine if this training system can significantly enhance their lumbopelvic control ability, reducing the risk of sports injuries caused by incorrect movements. Our ultimate goal is to enhance training effectiveness for players and promote a safer sports culture.

In addition, I would like to note that the interface shown in Figure 1 is not informative. There is no explanation of the matrix of figures (6) , and the interface is not translated into English (7)

Response:

We thank Reviewer 2 for the recommendations. According to the suggestions, we have added descriptions and changed Figure 1 to show how the system works in the revised manuscript. Moreover, we added Table 2 to show the function of the system. The revised contents are as follows:

(1)  In Figure 1 page 4 lines 124 to 127

Figure 1. The interface of the self-developed software. Panel (A) displays the real-time pelvic angle in anterior-posterior tilt, right-left tilt, and right-left rotation; Panel (B) sets limitations on pelvic angles during lumbopelvic control movements; Panel (C) buttons for controlling the application.

(2)  In Table 2 page 4 lines 129

Table 2. Function of the buttons on the interface

(3)  In 2. Materials and Methods 2.3. Real-Time Biofeedback System for Lumbopelvic Control page 4 lines 131 to 136

Central to our system is the integration of real-time biofeedback with lumbopelvic control training principles. Coaches and trainers can tailor pelvic angle constraints for various scenarios, offering athletes immediate visual feedback during exercises and issuing warnings if movements exceed established thresholds. The system tracks and visually represents pelvic tilt angles during exercises, empowering athletes to adjust their movements in real time for enhanced training precision.

Reviewer 3 Report

Comments and Suggestions for Authors

By conducting a study on 12 baseball players, this research collected data and compared the pelvic Angle detected by the real-time biofeedback system with the three-dimensional motion capture system. The comparison was made during the participants' side-sliding and single-leg squat exercises, and it revealed a medium-strong correlation between the two systems. This finding underscores the real-time biofeedback system's ability to accurately provide the correct pelvic Angle during the training process, thereby enhancing the training effect. The real-time biofeedback and adjustment methods proposed in this paper hold significant potential for improving the training effect. However, the data connection between the 3D motion capture system and the experimental object is not explained, and the specific evaluation performance index is not given。 Additionally, there are several typos/errors in the equations that need to be corrected before the final submission.

1.     The paper’s abstract should give the purpose, method, experimental conclusion, etc.

2. The author should give the principle of sample quantity selection, including at least 10 samples, and whether there is relevant literature. Does the selection of 12 samples meet the requirements of the experiment?

3. If only biological information is given in the sample object, does occupational information need to be supplemented?

4. A real-time biofeedback system adopts an inertial measurement unit. How do we carry out data acquisition? What motion information needs to be collected?

5. The use of the real-time biofeedback system in pelvic control is a promising approach. However, further discussion is needed on how to achieve real-time feedback with the combination of principles and how to adjust the training principles after information feedback. This aspect warrants the author's attention and could significantly enhance the effectiveness of the training principles.

6. How do you detect the posture information of the waist and pelvis? Can full attitude detection be achieved, as shown in Figure 2?

Author Response

Comments from Reviewer 3:

By conducting a study on 12 baseball players, this research collected data and compared the pelvic Angle detected by the real-time biofeedback system with the three-dimensional motion capture system. The comparison was made during the participants' side-sliding and single-leg squat exercises, and it revealed a medium-strong correlation between the two systems. This finding underscores the real-time biofeedback system's ability to accurately provide the correct pelvic Angle during the training process, thereby enhancing the training effect. The real-time biofeedback and adjustment methods proposed in this paper hold significant potential for improving the training effect. However, the data connection between the 3D motion capture system and the experimental object is not explained, and the specific evaluation performance index is not given。 Additionally, there are several typos/errors in the equations that need to be corrected before the final submission.

  • The paper’s abstract should give the purpose, method, experimental conclusion, etc.

Response:

We thank Reviewer 3 for the recommendations. According to the suggestions, since the original abstract already included the method and experimental conclusion, we have added a description of the purpose to the abstract of the revised manuscript. The revised contents are as follows:

(1)    In Abstract page 1, lines 15 to 16

Purpose: this study was to investigate the validity and reliability of the real-time biofeedback system for lumbopelvic control training

  • The author should give the principle of sample quantity selection, including at least 10 samples, and whether there is relevant literature. Does the selection of 12 samples meet the requirements of the experiment?

Response:

We thank Reviewer 3 for the recommendations. The initial pilot data included 6 participants. GPower 3.1 software was used to conduct a power analysis, determining the required sample size. The sample size of 10 was calculated based on the pilot data. The parameters used for calculation were a target power of 0.80, an alpha level of 0.05, and an effect size of 0.66. This study has received data from 12 participants, which exceeds the calculated initial sample size of 10. Therefore, 12 samples meet the requirements of the experiment. The revised contents are as follows:

(1)    In 2. Materials and Methods 2.1. Participants page 2, lines 82 to 84

To determine the appropriate sample size for our study, we utilized GPower 3.1 software. The initial pilot data included 6 participants. The sample size of 10 was calculated based on the pilot data. The parameters used for calculation were a target power of 0.80, an alpha level of 0.05, and an effect size of 0.66.”

  • If only biological information is given in the sample object, does occupational information need to be supplemented?

Response:

We thank Reviewer 3 for the recommendations. Training is based solely on biological information, and occupational information has not been utilized. This functionality can be added in future research to enhance the completeness of the system.

  • A real-time biofeedback system adopts an inertial measurement unit. How do we carry out data acquisition? What motion information needs to be collected?

Response:

We thank Reviewer 3 for the recommendations The real-time biofeedback system utilizing an inertial measurement unit (IMU) for lumbopelvic control training conducts data acquisition through a C++ back-end. The IMU captures pelvic angle data, including anterior-posterior tilt (AP), upward-downward obliquity (UD), and internal-external rotation (IE) values. These motion parameters are essential for analyzing and providing real-time feedback during the training sessions. The revised contents are as follows:

  • In 2. Materials and Methods 2.3. Real-Time Biofeedback System for Lumbopelvic Control page 3, lines 121 to 125

The real-time biofeedback system using the Xsens Awinda IMU for lumbopelvic control training conducts data acquisition through the C++ back-end. The IMU captures pelvic angle data, including anterior-posterior tilt (AP), upward-downward obliquity (UD), and internal-external rotation (IE) values. These motion parameters are crucial for analyzing and providing real-time feedback during training sessions.

  • The use of the real-time biofeedback system in pelvic control is a promising approach. However, further discussion is needed on how to achieve real-time feedback with the combination of principles and how to adjust the training principles after information feedback. This aspect warrants the author's attention and could significantly enhance the effectiveness of the training principles.

Response:

We thank Reviewer 3 for the recommendations. The angle range that needs to be achieved during the training is set according to the trainer's requirements. If the angle during training exceeds the set range, a warning will be displayed, and players must adjust their movements to the correct ones based on real-time feedback values. The revised contents are as follows:

  • In 5. Conclusions page 12, lines 341 to 351

This study emphasizes the potential impact of a real-time biofeedback system on lumbar pelvic control training for baseball players. Our system provides immediate feedback during exercises designed to enhance lumbopelvic control, significantly contributing to improved training outcomes. In our upcoming research, we plan to leverage this technology to enhance athletes' lumbar pelvic motion control, aiming to assess its effectiveness in reducing the risk of sports injuries resulting from incorrect movements. Our overarching goal is to not only optimize training effectiveness for players but also cultivate a safer sports culture. To achieve this, we have implemented an angle range within the training regimen, tailored to meet the specific requirements set by trainers. If the angle surpasses this predefined range during training, a warning prompt is displayed, prompting players to adjust their movements in real-time based on feedback values.”

  • How do you detect the posture information of the waist and pelvis? Can full attitude detection be achieved, as shown in Figure 2?

Response:

We thank Reviewer 3 for the recommendations. The placement of the IMU in Figure 2 is based on past literature, where participants securely attach an IMU to an elastic waist belt that covers the anterior superior iliac spines (ASIS) and posterior superior iliac spines (PSIS). The elastic waist belt covers the entire pelvic region, allowing for the detection of pelvic movements. The revised contents are as follows:

  • In 2. Materials and Methods 2.3. Real-Time Biofeedback System for Lumbopelvic Control page 3, lines 146 to 147

The elastic waist belt covers the entire pelvic region, allowing for the detection of pel-vic movements.”

Round 2

Reviewer 1 Report

Comments and Suggestions for Authors

From perspective of the product and practical clinical application value, the lack of the number of subjects was striking and the experimental data collection time seemed insufficient. Too many similar researches could be found respect to real-time biofeedback, while no impressive innovation was proposed in this manuscript.

Comments on the Quality of English Language

No suggestion.

Author Response

Comments from Reviewer 1:

From perspective of the product and practical clinical application value, the lack of the number of subjects was striking and the experimental data collection time seemed insufficient. Too many similar researches could be found respect to real-time biofeedback, while no impressive innovation was proposed in this manuscript.

Response:

Thank you for your valuable feedback. We appreciate your perspective on the product and its practical clinical application. Regarding the number of subjects and the experimental data collection time, we acknowledge the importance of these aspects and will consider ways to address them in future studies. As for the need for more impressive innovation, we will strive to introduce novel contributions and enhancements in our future work to make our research stand out in real-time biofeedback. Your insights are instrumental in guiding our future research directions.

Reviewer 2 Report

Comments and Suggestions for Authors

all comments have been corrected

Author Response

Comments from Reviewer 2:

all comments have been corrected 

Response:

We thank Reviewer 2 for the recommendations. We have carefully considered your suggestions and are grateful for the valuable insights they have provided.
